# Apicobasal Surfaceome Architecture Encodes for Polarized Epithelial Functionality and Depends on Tumor Suppressor PTEN

**DOI:** 10.3390/ijms232416193

**Published:** 2022-12-19

**Authors:** Anika Koetemann, Bernd Wollscheid

**Affiliations:** 1Department of Health Sciences and Technology, Institute of Translational Medicine, ETH Zurich, 8049 Zurich, Switzerland; 2Swiss Institute of Bioinformatics (SIB), 1015 Lausanne, Switzerland

**Keywords:** epithelial polarity, apicobasal, apical membrane, basolateral membrane, cell surface, sorting, polarized trafficking, PTEN, epithelial–mesenchymal transition, collective cell migration

## Abstract

The loss of apicobasal polarity during the epithelial-to-mesenchymal transition (EMT) is a hallmark of cancer and metastasis. The key feature of this polarity in epithelial cells is the subdivision of the plasma membrane into apical and basolateral domains, with each orchestrating specific intra- and extracellular functions. Epithelial transport and signaling capacities are thought to be determined largely by the quality, quantity, and nanoscale organization of proteins residing in these membrane domains, the apicobasal surfaceomes. Despite its implications for cancer, drug uptake, and infection, our current knowledge of how the polarized surfaceome is organized and maintained is limited. Here, we used chemoproteomic surfaceome scanning to establish proteotype maps of apicobasal surfaceomes and reveal quantitative distributions of, i.e., surface proteases, phosphatases, and tetraspanins as potential key regulators of polarized cell functionality. We show further that the tumor suppressor PTEN regulates polarized surfaceome architecture and uncover a potential role in collective cell migration. Our differential surfaceome analysis provides a molecular framework to elucidate polarized protein networks regulating epithelial functions and PTEN-associated cancer progression.

## 1. Introduction

In epithelial tissues, such as the respiratory epithelium and the blood–brain barrier, the plasma membrane has a polarized structure. The resulting apical and basolateral membrane domains act as gateways for signaling molecules and nutrients and are the interfaces for host–pathogen interactions and drug uptake. Remarkably, the functionality and responsiveness of epithelia are specific to each surface. This is exemplified by EGF-mediated signaling, which has long been known to differ upon apical versus basolateral induction [1,2,3]. Presumably, the proteins residing in each membrane domain—here collectively referred to as the apicobasal surfaceome—are decisive for side-specific epithelial operations, and disturbance of their polarized organization leads to altered functioning of the cell. Loss of apicobasal polarity is commonly observed in carcinogenesis and occurs during the epithelial-to-mesenchymal transition (EMT), in which epithelial cells dedifferentiate into the migratory mesenchymal phenotype that promotes metastasis [4]. Accordingly, expression aberrations of polarity-controlling proteins are observed in many different cancers [5,6]. Yet, we only have limited knowledge on polarized surfaceome organization and maintenance.

This can be partially attributed to the fact that, until recently, studies on polarized protein localization and trafficking have been largely based on confocal microscopy and limited to single model proteins [7]. A few studies have employed mass spectrometry (MS) for protein identification in the polarized membrane but have focused on the more easily accessible apical surface [8,9] or disregarded the quantities of proteins [10], a parameter that critically impacts the functional capacity of a membrane. Importantly, proteins do not act on their own in order to fulfill cellular functions but in stable or transient interaction with other proteins [11]. The composition of the proteome and its organization into functional modules through a plethora of protein–protein interactions are defined as the proteotype of a cell [12]. Knowledge of the proteotype of the polarized surfaceome is key to understanding the differential functioning of apical and basolateral membranes as a basis to resolve mechanisms of pathogenesis [13].

Chemoproteomic technology, in combination with quantitative MS analysis, enables the enrichment and reliable quantification of cell surface proteins [12,14,15]. In this study, we used quantitative chemoproteomic approaches for a detailed characterization of apical and basolateral proteotypes in order to identify key elements of their polarized regulation and explore how the synergy of local proteins translates into site-specific membrane functionality. In addition, we investigated the perturbation of apicobasal polarity, studying the impact of the tumor suppressor PTEN on the polarized surfaceome as well as on the epithelial proteotype in the context of carcinogenesis.

## 2. Results

### 2.1. Chemoproteomic Approach for Mapping of Polarized Surfaceomes

In order to map the apicobasal surfaceome quantitatively, we used filter-grown Madin–Darby canine kidney (MDCK) cells as the best-established in vitro system for epithelial polarity [16] in combination with chemoproteomics and stable isotope labeling of amino acids in cell culture (SILAC) (Figure 1A) [17,18]. Briefly, 6-day-old cultures of filter-grown MDCK cells allowed for specific labeling of apical and basolateral surface proteins with amine-reactive N-hydroxy-succinimide (NHS), which we confirmed by confocal microscopy (Figure 1B). We employed sulfo-NHS-SS-biotin to specifically label and enrich apical and basolateral proteins for subsequent MS analysis. To account for the presumably different total protein content of the two membrane domains, we used SILAC cultures to derive apical and basolateral proteins from the same cell growth area and to relatively quantify them by MS analysis based on the ratio of their heavy and light isotopes (Figure 1A). Final normalization for technical variations of heavy and light intensities was accomplished based on the overall heavy-to-light protein ratio measured in the intracellular fraction of each sample. Finally, identified proteins were filtered for evidence for localization at the plasma membrane using UniprotKB and the Surfy surfaceome predictor [19] (Appendix A).

### 2.2. Quantitative Protein Distribution Is the Key Feature of the Apicobasal Surfaceome

Using this approach, we were able to confidently quantify about 400 proteins across the polarized membrane of MDCK cells (Appendix A). From these data, we generated a quantitative map of the apicobasal surfaceome to characterize key features and the functional capacity of the two membrane domains (Figure 2A and Appendix A). Interestingly, most of the proteins were found in both surfaceomes. Only a few proteins (around 5%) were exclusively detected to be apical or basolateral, including several proteins for which the polarized localization has been previously described, as well as a number of novel apical or basolateral markers (Figure 3A). Intriguingly, although most surface proteins were generally detected in both membrane domains, more than 60% of them showed a quantitatively polarized distribution (apical:basolateral ratio <40:60 or >60:40). Notably, proteins with a distribution polarized towards the apical surface were substantially less frequent than towards the basolateral surface (average apical:basolateral ratio about 40:60), presumably due to the larger basolateral surface area (Figure 2B).

We exploited our data to resolve apparent contradictions in the field of apicobasal polarity and trafficking. Protein delivery to each membrane domain requires polarized intracellular sorting machinery and has been shown to be determined by sorting signals within cargo proteins that are recognized by specific adaptor proteins [20,21]. Despite the proposed function of glycosylphosphatidylinositol (GPI) anchors as apical sorting signals [16,22], a previous MS-based study on MDCK cells suggested a non-polarized distribution of the great majority of GPI-anchored proteins [23]. The presented quantitative data indicate that although most GPI-anchored proteins are present in both domains, 86% of the detected GPI-anchored proteins do show a polarized distribution—in a quantitative fashion—with a clear tendency toward the apical surface (Figure 2H).

#### 2.2.1. Protein Abundances Indicate Functional Capacities of Apical versus Basolateral Membranes

Differences in protein abundance in the apical versus the basolateral membrane are indicative of distinct functions of these membrane domains. For instance, the basolateral surface appears to predominantly mediate cell adhesion and functions involved in the cellular community, whereas both apical and basolateral membranes are suggested to fulfill various tasks of the immune and inflammatory system (Figure 2A). More precisely, we found strongly polarized localizations for different receptors of the immune response, transmembrane transporters, and G-protein-coupled receptors (Figure 2G and Appendix A). Based on receptor abundance, we can deduce that the detection of bacterial lipopolysaccharide via CD14 as well as SCARB1-dependent uptake of lipids and HDL occurs exclusively at the apical cell surface. Further, response to proinflammatory interleukin-1 via the IL1R1 receptor is expected to be mainly mediated through the apical membrane. In contrast, planar polarity (via Wnt signaling) is suggested to be regulated through the basolateral surface, based on the localization of frizzled receptors 1 and 5. In addition, pro-angiogenic signaling of the PTGER2 receptor in response to the prostaglandin E2 hormone, a pathway currently discussed as a potential drug target for cancer therapy, is only expected from the basolateral side. Conclusively, the quantitative map of the apicobasal surfaceome enables us to identify cellular functions that are orchestrated by epithelia in a polarized fashion.

#### 2.2.2. Polarized Distribution of Modulatory Proteins

Whereas various receptor tyrosine kinases (RTKs) that mediate major cellular functions such as growth signaling (e.g., EGFR and MET) showed rather similar apical and basolateral abundances (Figure 2C), many of their modulatory interaction partners were found to be strongly polarized. In particular, about three-fourths of receptor-type tyrosine–protein phosphatases (PTPRs/PTPNs) were detected with substantial polarization, indicating side-specific as well as universal functions of these phosphatases (Figure 2D). PTPRs have been shown to dephosphorylate specific sites of different RTKs and thereby have a direct impact on downstream signaling [24,25]. Proteins of the tetraspanin family, which are believed to regulate the function of their interaction partners by membrane compartmentalization [26], also showed a polarized distribution, mostly with a strong preference for the basolateral domain (Figure 2F). Furthermore, our quantification revealed the polarized distribution of several proteolytic enzymes that are required for the shedding of signaling molecules from the cell surface or protein processing in the extracellular space (Figure 2E). These findings highlight modulatory proteins that arrange, process, or modify other proteins as key elements of polarized membrane organization.

#### 2.2.3. Quantitative Protein Distributions Give Rise to Polarized Functional Networks

Subcellular compartmentalization is a key regulator of the proteotype, as it generates local networks of specific protein subsets [27,28]. Accordingly, the quantitative distribution of proteins across the apicobasal surfaceome determines their interaction space within each membrane domain, which may result in differential regulation of apical and basolateral functions. In the context of growth signaling, for instance, we know that apical EGF stimulation induces a different intracellular response than basolateral stimulation [1,2,3]. However, we show here that both the EGF precursor and its receptor EGFR are relatively evenly distributed across the polarized membrane (with an apicobasal ratio of 50:50 and 40:60, respectively). Therefore, we created an in silico map of the apicobasal protein interaction network exemplified by the EGF/EGFR system (Figure 3B) and examined how the differential abundance of known interaction partners in the apical versus basolateral membrane may affect the functioning of this signaling machinery.

Firstly, we found the ADAM10 protease that is known to cleave the EGF precursor from the cell surface to release the active growth factor, mainly in the basolateral domain. In contrast, the CPM peptidase, which is believed to control growth factor activity by hydrolysis, predominantly localizes to the apical surface. This suggests that EGF of epithelial origin exhibits mostly basolateral activity under physiological conditions, despite the unpolarized distribution of the precursor.

Furthermore, we found a number of RTK co-receptors and phosphatases, previously shown to modulate the phosphorylation and function of the EGF receptor, to be quantitatively polarized. Unlike EGFR (ErbB1), other members of the ErbB growth factor receptor family (ErbB2-4) were detected as differentially abundant in the apical versus the basolateral membrane domain. ErbB receptors are known to form homo- and heterodimers upon binding of particular ligands and induce alternative cross-phosphorylation and intracellular signaling pathways depending on the dimer formed [29]. In addition, several PTPR phosphatases that are known to interact with EGFR showed a polarized distribution. Based on protein abundance, the formation of EGFR/EGFR homodimers and the regulation by PTPRJ are more likely to occur in the apical domain, whereas EGFR/ErbB3 heterodimers and interaction with PTPRS and PTPRG may have a higher prevalence in the basolateral domain. This distinct interaction potential with other RTKs and phosphatases suggests that differentially phosphorylated proteoforms of EGFR are generated upon apical versus basolateral EGF binding to modulate downstream signaling.

Additionally, apicobasal EGF signaling may be influenced by crosstalk with other receptor signaling pathways and tetraspanins, such as CD9 and CD82, which are thought to determine the substrate specificity of ADAM proteases. Conclusively, cellular responsiveness to, e.g., growth factors may be tailored to the specific functions of the apical and basolateral surface by distinct interaction networks that are dominated by polarized distributions of modulatory proteins.

### 2.3. Tracking of Polarized Protein Trafficking Suggests Intertwining of Sorting Routes

The current knowledge on mechanisms of polarized protein sorting does not reflect the diverse quantitative protein distributions we observed. Therefore, we explored the applicability of chemoproteomic approaches to directly track polarized protein delivery. We employed pulsed SILAC in combination with a membrane-fixation strategy based on tannic acid. Tannic acid has been shown previously to block the fusion of intracellular vesicles with the plasma membrane and to fix the apical or basolateral membrane of filter-grown MDCK cells in a side-specific manner [7]. We confirmed the full integrity of treated cell layers by confocal microscopy (Figure 4A). After tannic acid treatment to block vesicular delivery to one membrane domain, we switched the amino acid source to heavy isotopes, allowing us to track the trafficking of newly synthesized (heavy) proteins to the other membrane domain using cell surface proteomics (Figure 4B,C; roughly 120 proteins quantified for each domain). Four hours after blocking delivery to the basolateral membrane, we observed that the heavy version of most proteins accumulated in the apical membrane compared with the control that was not treated with tannic acid (Figure 4B). In contrast, blocking apical protein delivery had very little effect on protein delivery to the basolateral surface, except for, e.g., the transcytotic LDL receptor (Figure 4C). Hence, most proteins in the biosynthetic pathway were redirected to the apical surface if basolateral delivery was denied but not the other way around. Interestingly, we also found that the turnover of a few surface proteins was location-specific, with more rapid incorporation of the heavy version in one membrane domain compared with the other (Figure 4D). In conclusion, this experiment demonstrated that chemoproteomic approaches are also potent tools to dissect polarized protein trafficking and showed an intertwining of apical pathways with basolateral routes.

#### 2.3.1. Impaired PTEN Function Causes Alterations in the Polarized Surfaceome

After establishing the quantitative composition of the apicobasal surfaceome in a physiological setting, we aimed to understand polarity-associated defects that may contribute to cancer progression. PTEN is one of the most frequently mutated tumor suppressors in humans. Besides its established role as an antagonist of PI3-kinase-dependent growth signaling, PTEN has also been shown to play a role in EMT [30] and to be a determinant of epithelial polarity. PTEN functions as a protein and lipid phosphatase known to, e.g., dephosphorylate phosphatidylinositol (3,4,5)-trisphosphate (PtdIns(3,4,5)P3) to phosphatidylinositol (4,5)-biphosphate (PtdIns(4,5)P2). It has been shown that PtdIns(3,4,5)P3 plays a role in establishing the basolateral domain and is excluded from the apical membrane, whereas PtdIns(4,5)P2 and PTEN localize to the apical membrane in MDCK cyst cultures. This localization is required for the recruitment of the apical polarity complex, which coordinates the formation of tight junctions essential for polarity [31,32]. We hypothesized that PTEN function is also crucial for the polarized localization of cell surface proteins.

In order to investigate the effect of the phosphatase activity of PTEN on the apicobasal surfaceome composition, we treated filter-grown MDCK cells with the small-molecule inhibitor SF1670 at a low concentration (1 μM versus IC50 = 2 μM to retain target specificity). Using short incubations (4 h), we initially targeted the acute response of polarized protein trafficking, while avoiding cellular contra-regulation on the level of protein expression. Importantly, the treatment had no impact on either the integrity of the cell layer (Figure 5A) or the cellular abundance of most proteins under these conditions (Figure 6A). In a label-free MS experiment, we found that the quantities of a number of apical surface proteins were altered upon PTEN inhibition compared with the mock-treated control, although most proteins remained unchanged. Among the proteins with increased abundance were receptors that regulate cellular growth and cohesion, whereas a few proteins involved in apicobasal polarity and transport to the cell surface were downregulated (Figure 5B). This effect was rescued by simultaneous inhibition of the PI3-kinase (Figure 5C), indicating a PtdIns phosphorylation-dependent trafficking mechanism of altered apical surface proteins.

In line with the apical localization of PTEN, the basolateral surfaceome was unaffected by PTEN inhibition after 4 h. However, at a later stage of PTEN inhibition (12 h), we observed substantial changes in the basolateral surfaceome composition (Figure 5D). For instance, we detected enhanced surface (but not overall) expression of growth factor receptors such as EGFR, MET, IGF1R, and ERBBs. Intriguingly, Gene Ontology analysis of proteins with an elevated abundance revealed an enrichment of proteins that mediate cell adhesion and migration, suggesting that cells gained motility (Figure 5E). At the same time, increased expression of cell–cell adhesion proteins, including epithelial markers such as CDH1/E-cadherin, indicated enhanced cell cohesion. Among the downregulated proteins were, for instance, regulators of extracellular matrix organization. Strikingly, a number of proteins that are known to play important roles in cancer progression and invasion, such as Podocalyxin and inhibitory Calumenin, were found upon PTEN inhibition with increased or decreased basolateral levels, respectively. Taken together, these data suggest that the tumor suppressor PTEN does not only regulate apical surfaceome maintenance but ultimately also cellular motility through modulation of the basolateral surfaceome.

#### 2.3.2. Impaired PTEN Function Causes Massive Proteotype Reorganization

In order to resolve the underlying mechanisms of surfaceome changes following the loss of PTEN functionality, we performed proteotype analyses of polarized MDCK cells in a time course of inhibitor treatment (Figure 6). In a label-free MS experiment, we identified about 4260 proteins in total. After 4 h of inhibitor treatment, we only observed abundance changes in a few proteins (Figure 6A), whereas at a later stage (8 h) the inhibitor treatment led to quantitative alterations in more than 60 proteins (Figure 6B). After 12 h of PTEN inhibition, profound expression changes, comprising roughly one-fifth of the detected proteins, indicated a massive reorganization of the cellular proteotype (Figure 6C).

The earliest regulated proteins that also persisted throughout the time course and are thus potential initiators of the observed effects were RHOC, TMEM201, and a HMGN protein (2, 3, or 4). RHOC and TMEM201 are known to regulate cell junction assembly and cell migration (Gene Ontology), in line with the subsequent basolateral expression of the respective proteins described above. HMGN proteins bind to nucleosomes for transcriptional regulation, with the downstream effects being unclear.

At the intermediate time point of 8 h, we identified several regulators of gene transcription and RNA processing/degradation as well as known regulators of migration that might be driving proteotype reorganization (Figure 6D). Additionally, we found components of the methylosome complex among upregulated proteins, which mediates the arginine methylation of histones and other protein targets. The methyltransferases PRMT1 and PRMT5 have been shown to be overexpressed in many cancers and to regulate cancer cell migration and invasion [33,34,35]. Transcriptional regulation by histone methylation may therefore play a crucial role for the subsequent effects of PTEN inhibition.

Among the proteins with elevated levels after 12 h of inhibitor treatment were proteins involved in RNA processing and protein translation as well as chaperones (e.g., HSP90) and proteins associated with extracellular exosomes. Importantly, we did not detect known markers of EMT or mesenchymal differentiation at any point, in agreement with sustained expression of cell–cell contacts. However, we found a number of established drivers of cancer progression and invasion (e.g., mTOR, STAT1/5, RHOC, ARF6, HMGB, and CRK) with increased expression levels as well as known suppressors (e.g., AIFL1 and SLIT3) with decreased expression levels upon PTEN inhibition.

Taken together, our data show that impaired PTEN function does not lead to loss of epithelial cell polarity but to a global reorganization of the polarized surfaceome as well as the intracellular proteotype that resembles a migratory but not mesenchymal phenotype.

## 3. Discussion

In this study, we used chemoproteomic approaches to map the apicobasal surfaceome in an in vitro model and to investigate perturbations associated with carcinogenesis. We specifically identified quantitative differences in protein composition as a major factor for the epithelial surface organization. This was particularly the case for modulatory proteins that organize, process, or modify other proteins in order to regulate their activity. A recent study used a similar approach to quantify apicobasal protein distributions in MDCK cells for research on polarized protein sorting [36]. The data are remarkably consistent with our map of the apicobasal surfaceome, confirming our results by mass spectrometry as well as immunoblotting (e.g., PODXL = 93 ± 1% vs. 99 ± 1% apical, CEACAM1 = 96 ± 3% vs. 93 ± 5% apical, CD44 = 22 ± 1% vs. 20 ± 17% apical, MET = 41 ± 2% vs. 27 ± 30% apical in this study vs. the study by Caceres et al., respectively, with a median difference of 14.4% for the MS quantification of 230 proteins detected in both studies). The two studies establish quantitative chemoproteomics as a powerful tool to investigate epithelial polarity.

Differential protein abundances suggest that there are distinctive interaction potentials for each protein within the apical versus the basolateral domain. This is likely to influence the nanoscale organization of proteins, perhaps with modulatory proteins and co-receptors, which is believed to be crucial for physiological cell signaling [13]. Differential apicobasal EGF signaling has been associated with domain-specific EGFR binding of intracellular signaling proteins [1]. We hypothesize that the upstream molecular basis for such observations is a polarized receptor nano-scale organization resulting in the generation of side-specific (phospho) proteoforms that recruit different downstream signaling proteins. Future studies may follow up on the emergence of specific apical and basolateral phosphoforms during signal transduction and verify domain-specific interactions with regulatory proteins by proximity-labeling or other strategies [37]. It will also be interesting to investigate how, for instance, distinct glycoforms or additional mechanisms of membrane compartmentalization (e.g., cilia) may contribute to apical and basolateral functionality. Moreover, the surfaceome map could be extended by the incorporation of additional types of data (e.g., RNA-seq data from polarized cells to identify splice variants).

Our data on quantitative surfaceome polarity emphasize the complexity of polarized protein trafficking. The occurrence of known apical and basolateral sorting signals is usually not limited to one membrane, and it has been shown previously that apicobasal sorting of a particular protein can depend on multiple sorting signals [38,39]. In this study, we showed that most proteins can be redirected to the apical surface if basolateral delivery is inhibited, indicating an intertwining of apicobasal trafficking routes. Therefore, we propose a model in which apical sorting machineries act competitively with the basolateral (default) route in order to establish a specific protein distribution. Competitive sorting might be a beneficial mechanism to enable the “flexible phenotype” of epithelia in distinct tissues and physiological conditions. In-depth bioinformatic analyses of the data generated in this work may identify patterns in apical versus basolateral protein features required for such a sorting system.

Our findings concerning the tumor suppressor PTEN demonstrate that PTEN is implicated in apical localization of cell surface proteins, in line with previous studies of PTEN functions in polarity [31,32]. Further, we provide evidence that inhibited PTEN activity causes increased expression of known drivers of cancer cell migration and invasion, followed by a substantial reorganization of the epithelial proteotype as well as the basolateral surfaceome that suggests cell motility. While these findings draw parallels to EMT, we did not find any evidence for a loss of cell cohesion and polarity or alterations of major EMT markers. This phenotype may resemble collective cell migration, describing the movement of cell sheets rather than single cells, which has been shown to play a role in wound healing as well as tumor metastasis [40]. PTEN has been shown previously to play a role in leukocyte motility and in restricting cell migration in wound healing [41,42,43]. In fact, E-cadherin—as a typical marker lost during EMT that we found to be upregulated upon PTEN inhibition—has been shown to be essential for collective cell invasion [44]. Interestingly, higher concentrations of PTEN inhibitor led to cell detachment in large patches (personal observation). Collectively, our findings point towards an acquired motility, potentially collective cell migration, upon PTEN inhibition and identify several potential driver proteins thereof. Future studies may validate these findings by PTEN knock-down experiments and follow up on them by detailed phenotyping of PTEN-impaired epithelia, including, e.g., migration assays, exosome analysis, and profiling of histone methylation.

Overall, the map of the apicobasal surfaceome generated here enabled system-wide insights into the protein organization and maintenance required for the polarized operation of epithelial cells. Providing concepts of epithelial plasticity and the role of the tumor suppressor PTEN, our findings can facilitate future studies of apical and basolateral functions and mechanisms of cancer progression.

## 4. Materials and Methods

### 4.1. Chemicals

All chemicals and cell culture reagents were from Sigma-Aldrich unless stated otherwise.

### 4.2. Cell Lines

MDCK II cells were a gift from Yohei Yamauchi (University of Bristol).

### 4.3. Mammalian Cell Culture, SILAC Labeling, and Polarized Culture

Cells were grown at 37 °C and 5% ambient CO_2_ to approximately 80% confluence in 140 × 20 mm dishes (Nunclon^TM^ Delta Airvent, Thermo Fisher, Waltham, MA, USA) with Dulbecco’s Modified Eagle’s Medium, high glucose, GlutaMAX^TM^ supplement (DMEM, Thermo Scientific, Waltham, MA, USA), 10% fetal bovine serum (FBS, BioConcept, Allschil, Switzerland), and 1% penicillin–streptomycin. For SILAC labeling, cells were grown in SILAC DMEM high glucose with stable glutamine and without arginine and lysine (Pan Biotech, Aidenbach, Germany), supplemented with 15% dialyzed FBS (Pan Biotech, Aidenbach, Germany), 1% penicillin–streptomycin, 10 mM HEPES (Gibco), non-essential amino acids (Gibco), 200 mg/L L-proline, 42 mg/L L-arginine or L-arginine [10], and 73 mg/L L-lysine or L-lysine [8] (Silantes, München, Germany) for 7 days, and the incorporation rate of heavy amino acids was determined to be 95–98% by MS analysis. For polarized cultures, 5 × 10^6^ or 0.5 × 10^6^ cells, respectively, were seeded into 75 mm or 12 mm polycarbonate Transwells^TM^ with a pore size of 0.4 µm (Corning) and cultured for 6–7 days while changing the medium every second day.

### 4.4. Confocal Microscopy Imaging

To evaluate the tightness of filter-grown cell cultures, cells were labeled at the apical or basolateral surface using 1 mM sulfo-NHS-Cy5 (Lumiprobe) in PBS, pH 8.0 for 15 min on ice in the dark and washed three times with PBS before fixation. For all microscopy samples, cells were fixed in 4% paraformaldehyde for 10 min at room temperature and permeabilized with 0.1% TritonX-100 in blocking buffer (1% FBS, 1% BSA in PBS with 0.02% sodium azide) for 10 min at room temperature. Samples were blocked for 1 h at room temperature or overnight at 4 °C in a blocking buffer. Cells were subsequently incubated with mouse anti-ZO1-AF555 antibody (Thermo Invitrogen MA3-39100-A555, 1:100) for 1 h with shaking. Nuclei were stained with 1 µg/mL Hoechst (Molecular probes H1399), and F-actin was stained with phalloidin-iFluor488 (Abcam, 1:1000) for 15 min at room temperature. Samples were fixed with 4% paraformaldehyde for 10 min at room temperature and mounted with Prolong Gold Antifade reagent mounting medium (Molecular Probes). Images were taken with a Leica TCS SP2 confocal microscope and processed using the FIJI software.

### 4.5. Tannic Acid Membrane Fixation and Pulsed SILAC

Filter-grown SILAC-light cells were washed once with pre-warmed PBS and incubated with 0.5% tannic acid in pre-warmed PBS in the upper or lower chamber of the Transwell for 10 min at room temperature. Cells were then washed with PBS and incubated for 4 h with SILAC-heavy medium at 37 °C. Cell surface proteins of the apical or basolateral membrane domains were then biotinylated and enriched for MS analysis, as described in Section 4.7.

### 4.6. Inhibitor Treatments

Filter-grown cells were treated with 1 μM SF1670 (Lucerna Chem) and/or 1 μM LY29400 in 20 mL of serum-free DMEM, 1% penicillin–streptomycin per Transwell (8 mL apical; 12 mL basal) at 37 °C for the indicated incubation times, and the inhibitor solution was refreshed every 4 h (0.002% DMSO end concentration). In a control, cells were mock-treated with 0.002% DMSO. Cell surface proteins of the apical or basolateral membrane domains were then biotinylated and enriched for MS analysis as described in Section 4.7.

### 4.7. Cell Surface Protein Enrichment

For side-specific enrichment of apical and basolateral proteins, filter-grown MDCK cells were labeled with a sulfo-NHS-SS-biotin conjugate (Pierce™ Premium Grade, Thermo Scientific, Waltham, MA, USA). For this purpose, cultures were placed on ice and washed twice with ice-cold PBS. Cells were rinsed on the side to be labeled with cold PBS, pH 8.0, and the opposite chamber was filled with 25 mM Tris, pH 8.0. The apical or basolateral cell surface was then labeled with 1 mM sulfo-NHS-SS-biotin for 15 min on ice. Cells were rinsed on the labeled side with 25 mM Tris, washed with PBS, and harvested by scraping in PBS and centrifugation (5 min at 300 rpm and 4 °C).

Cells were lysed on ice in lysis buffer (50 mM NH_4_HCO_3_ buffer, 0.25% RapiGest, complete protease inhibitor cocktail (Roche)) by sonication (Dr. Hielscher sonicator, 30 s at 80% amplitude and 80% cycle time). For apical versus basolateral protein quantification, heavy and light SILAC cell lysates were pooled. For whole cell proteome analysis, a sample of the lysate was taken before surface protein enrichment. Biotinylated proteins were enriched on 100 µL streptavidin Ultralink beads (Thermo Scientific, Waltham, MA, USA) for 1.5 h at room temperature with rotation. Beads were pelleted by centrifugation (3 min at 300× *g*) and the supernatant was collected for analysis of the unbound fraction. Whole cell lysates and unbound fractions were further processed as described in Section 4.8. Beads were washed 5 times each with 1 mL of 5 M NaCl, StimLys buffer (137 mM NaCl, 50 mM Tris, pH 7.8, 150 mM glycerol, 0.5 mM EDTA, 0.1% Triton X-100), 80% isopropanol, 100 mM sodium bicarbonate, pH 11, and 50 mM NH_4_HCO_3_ buffer using Bio-Spin^®^ columns (BioRad, Hercules, CA, USA) in combination with a vacuum manifold. Washed beads were transferred to 0.5 mL Eppendorf tubes, and proteins were eluted in 250 µL of 50 mM NH_4_HCO_3_ buffer, 5 mM TCEP, and 0.1% Rapigest by rotation for 1 h at 37 °C. The eluate was collected and combined with an additional bead wash performed with 250 µL of 50 mM NH_4_HCO_3_ buffer.

### 4.8. Sample Preparation for MS Analysis

All protein samples were reduced with 5 mM TCEP for 30 min at room temperature (except for cell surface protein enriched samples, which were reduced during the elution step), alkylated with 5 mM iodoacetamide for 30 min in the dark, and digested overnight with trypsin (Promega; 1 µg for cell surface enrichment samples, 1:100 for lysate samples) at 37 °C and 300 rpm. Samples were acidified to pH 2 by the addition of formic acid (FA) and centrifuged (10 min at 16,000 rpm at room temperature). Peptides were desalted on a reverse-phase C18 column (Nest Group) and eluted with 50% acetonitrile (ACN), 0.1% FA. Solvent was evaporated using a SpeedVac^®^ Concentrator (Thermo Scientific, Waltham, MA, USA). Peptides were resuspended in 5% ACN, 0.1% FA, and iRT retention time peptides (Biognosys, Schlieren, Switzerland) were spiked in for subsequent LC-MS/MS analysis.

### 4.9. LC-MS/MS Analysis

Peptides were analyzed on an Orbitrap Fusion mass spectrometer (Thermo Scientific, Waltham, MA, USA) equipped with a nano-electrospray ion source (Thermo Scientific, Waltham, MA, USA) and coupled to a nano-flow high-pressure liquid chromatography (HPLC) pump with an autosampler (EASY-nLC II, Proxeon). Peptides were separated on a reversed-phase chromatography column (75 μm inner diameter PicoTip™ Emitter, New Objective) that was packed in-house with a C18 stationary phase (Reprosil Gold 120 C18 1.9 μm, Dr. Maisch). Peptides were loaded onto the column with 100% buffer A (99.9% H_2_O, 0.1% FA) at 800 bar and eluted at a constant flow rate of 300 nL/min with a gradient of buffer B (99.9% ACN, 0.1% FA) with a subsequent wash step at 90% buffer B. The peptide concentration in the samples was determined using a Nanodrop spectrophotometer (Thermo Scientific, Waltham, MA, USA) at 280 nm. For the analysis of cell surface protein enrichment samples and the corresponding unbound fractions, 1 µg of peptides was separated on a 15 cm column with a 90 min linear gradient of 5–35% B, followed by a 5 min gradient to 50% B and a 4 min gradient to 90% B. For the analysis of lysates, 3 µg of peptides was separated on a 45 cm column with a 200 min linear gradient of 5–35% B, followed by an 18 min gradient to 50% B and a 10 min gradient to 90% B. Between batches of runs, the column was cleaned with two steep consecutive gradients of ACN (10–98%).

The MS was operated in data-dependent acquisition mode with a cycle time of 3 s and using all of the parallelizable time for ion injection. High-resolution MS scans were acquired in the Orbitrap (120,000 resolution, automatic gain control target value 2 × 10^5^) within a mass range of 395 to 1500 *m*/*z*. Precursor ions were isolated in the quadrupole with an isolation window of 2 *m*/*z* and fragmented using higher-energy collisional dissociation to acquire MS/MS scans in the Orbitrap (30,000 resolution, intensity threshold 2.5 × 10^4^, target value 2 × 10^5^). Dynamic exclusion was set to 30 s. Instrument performance was evaluated on the basis of regular quality control measurements using a yeast lysate and the iRT retention time peptide kit (Biognosys).

### 4.10. Data Analysis and Visualization

Data were collected from at least three experimental replicates per condition (independent Transwell cultures on the same occasion), as indicated in the figure legends of the respective datasets. Mass spectrometry data were analyzed using the MaxQuant software (version 1.6.0.16) using default settings if not stated otherwise. Searches were performed against the canine UniProt FASTA database (June 2018) and a contaminant database with a false discovery rate (FDR) of 1% at the protein and peptide level. For the analysis of cell-surface samples, the NHS modification (C_5_O_2_SNH_7_ = 145.0197Da at K) was selected as an additional variable modification, which was also used for quantification, and three missed cleavages were allowed. For SILAC-based quantification, multiplicity was set to 2, with Arg10 and Lys8 as heavy labels. For label-free quantification (LFQ) of proteome samples, “MaxQuant LFQ” was used. The “match between runs” function was enabled in all analyses.

Bioinformatics analysis was performed with Perseus (version 1.6.0.7) and Microsoft Excel. In order to filter out probable intracellular contaminants in the cell-surface-enriched samples, a “canine surfaceome” list was established, containing all canine proteins annotated as “Cell membrane”, “GPI-anchor”, and “secreted” in UniProtKB as well as the predicted surface proteins contained in the bronze set of “Surfy” [19], translated from human to canine IDs (Appendix A). For apicobasal quantification, apical and basolateral fractions of detected proteins were calculated from heavy and light intensities and normalized to the overall heavy-to-light ratio measured in unbound fractions of the protein enrichment. Data were filtered for quantification in at least two out of four replicates and are presented as medians of all replicates. As a measure of the variation of the quantification across replicates, the median absolute deviation (MAD) was calculated. Proteins with a median value of 0 (only basolateral) or 1 (only apical) were only considered as exclusively apical and basolateral proteins, respectively, if MAD <10%. Pulsed SILAC data were filtered for at least two valid values in a group and log-transformed before performing statistical testing using a two-sided t-test and permutation-based FDR (Student’s *t*-test, FDR = 0.05, s0 = 0.1). For label-free surfaceome analyses, unfiltered MS intensities were median-normalized and filtered for at least 2 unique peptides and 5 MS/MS counts. Protein data were filtered for quantification in at least 2 replicates per condition and LFQ-intensities were log-transformed. The statistical significance of results was determined with a two-sided t-test and a permutation-based FDR calculation using the Perseus software for the generation of volcano plots (FDR = 0.05, s0 = 0.1). Functional annotations of proteins were made manually based on information from UniProtKB and Gene Ontology. Gene ontology enrichment analyses of biological processes were conducted against the surfaceome list as a reference and using Fisher’s exact test with Bonferroni correction. Known protein–protein interactions were extracted from the STRING database using a minimum score of 0.4, excluding text mining. A relative apicobasal interaction potential score was calculated based on the quantitative protein distribution using the following formula: log_2_ [(AP_Node1_ × AP_Node2_)/(BL_Node1_ × BL_Node2_)], with AP being the apical fraction and BL being the basolateral fraction of the total surface pool of a protein. Data were visualized using the Perseus platform, Cytoscape, BioVenn, and R.

## Figures and Tables

**Figure 1 ijms-23-16193-f001:**
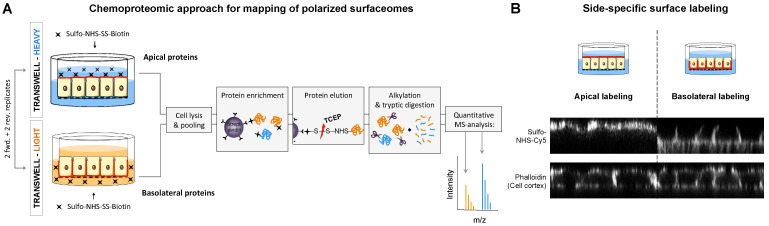
Chemoproteomic approach to map polarized protein distributions across the apicobasal surfaceome. (**A**) Workflow: MDCK cells are metabolically labeled with light and heavy isotopes of amino acids using SILAC and polarized in Transwell™ cultures. Apical and basolateral surface proteins are tagged in differentially labeled SILAC cultures using a sulfo-NHS-SS-biotin conjugate. Heavy and light cell lysates are mixed, and biotinylated surface proteins are enriched on streptavidin beads. Enriched proteins are eluted by reduction of the disulfide bond of the biotin conjugate, alkylated and digested with trypsin, and quantified by mass spectrometry. Proteins derived from the apical and basolateral surface are quantitatively compared based on their heavy-to-light ratio. Experiments were conducted in quadruplicates, including two forward and two reverse SILAC replicates. (**B**) Side-specific protein labeling with sulfo-NHS-Cy5 on the apical and basolateral surfaces of filter-grown MDCK cells was confirmed by confocal microscopy imaging. Phalloidin staining of F-actin was used to visualize the cell cortex as a reference.

**Figure 2 ijms-23-16193-f002:**
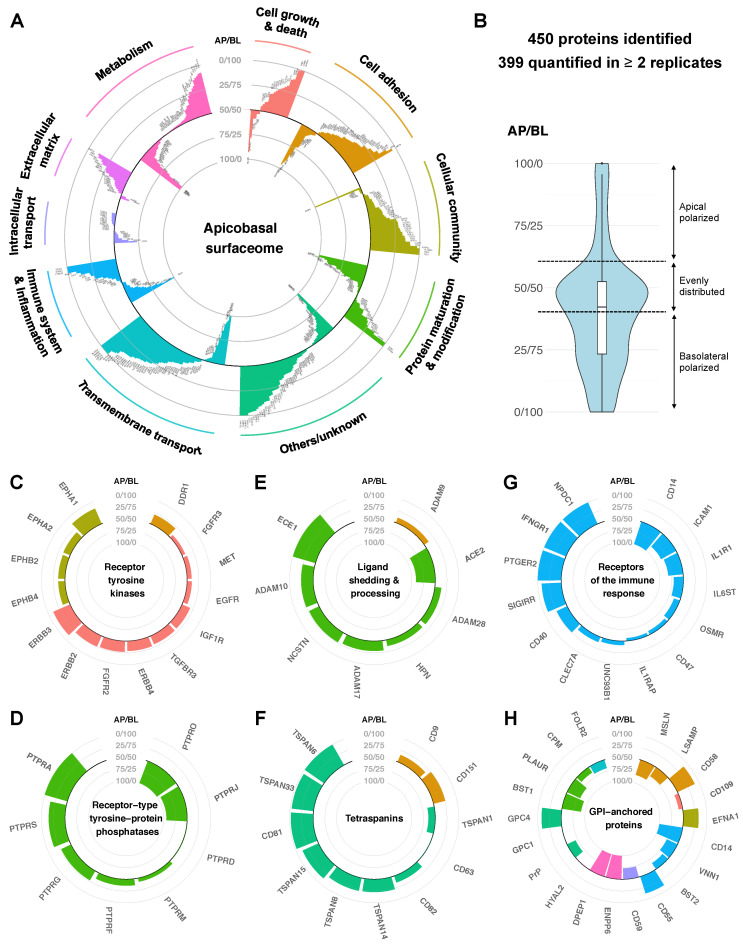
Quantitative apicobasal distribution (AP/BL ratio) of cell surface proteins on filter-grown MDCK cells based on SILAC-complemented chemoproteomics. (**A**) A quantitative map of the apicobasal surfaceome (apical tendencies more central, basolateral tendencies more peripheral in the plot). (**B**) Overall apicobasal protein distribution shows quantitative polarization for more than 60% of the detected proteins with an average ratio of 40:60. (**C–H**) Quantitative maps of different protein classes across the apicobasal surfaceome: (**C**) receptor–tyrosine kinases, (**D**) receptor-type tyrosine–protein phosphatases, (**E**) proteins involved in ligand shedding and processing, (**F**) tetraspanins, (**G**) receptors of the immune response, and (**H**) GPI-anchored proteins. Classification according to the Gene Ontology and KEGG pathway databases (*n* = 4, with 2 SILAC forward and 2 reverse experiments).

**Figure 3 ijms-23-16193-f003:**
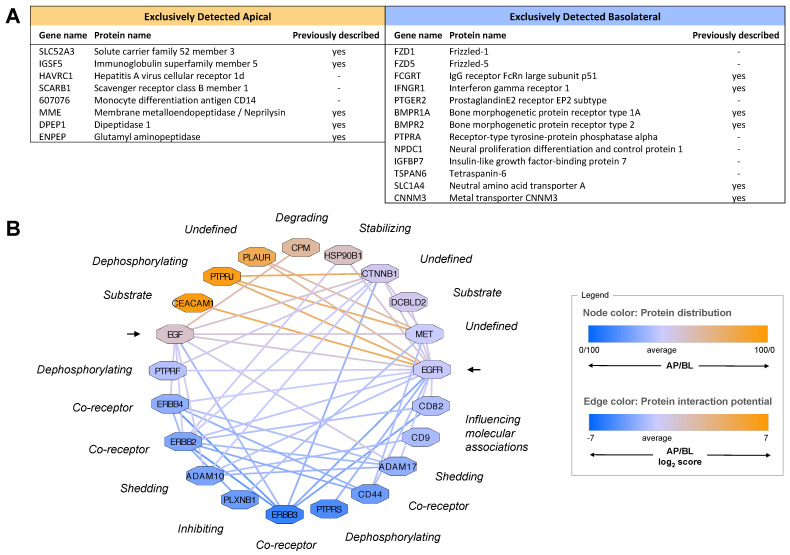
The apicobasal proteotype is determined by polarized protein quantities. (**A**) Only very few proteins were detected exclusively in the apical or basolateral membrane of filter-grown MDCK cells based on SILAC-complemented chemoproteomics (*n* = 4, with 2 SILAC forward and 2 reverse experiments). (**B**) Representation of the polarized interaction network influencing EGF growth signaling based on quantitative proportions of proteins within the apical versus basolateral membrane. Shown are selected known protein interactions and their functional impact on EGF and EGFR (arrows) according to the STRING and Uniprot databases as well as the literature. The quantitative distribution of protein nodes across the apicobasal surfaceome and the resulting likelihood of interaction within each membrane domain are color-coded (color scale from orange = apical to blue = basolateral).

**Figure 4 ijms-23-16193-f004:**
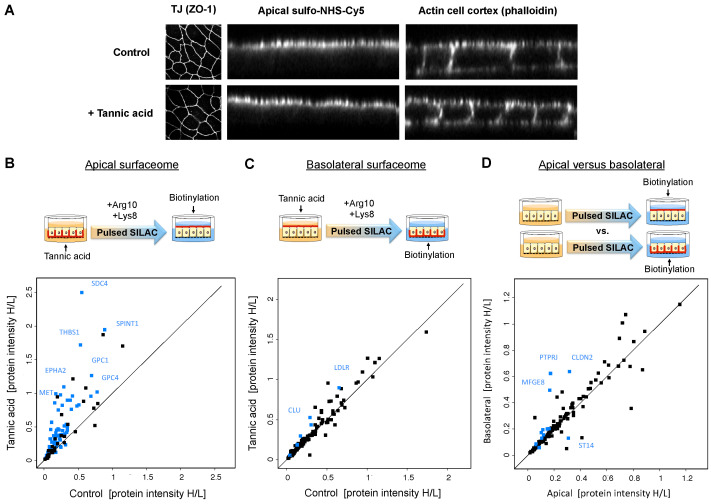
A pulsed SILAC experiment reveals flexibility in the delivery of newly synthesized proteins to the apical and basolateral surfaces upon side-specific blocking by membrane fixation using tannic acid. (**A**) Confocal microscopy imaging of filter-grown MDCK cells shows intact cell layer integrity 4 h after tannic acid treatment (0.5% for 10 min) compared with the untreated control, with no effect on the expression of the tight junction (TJ) protein ZO-1 (left panel) nor on the side-specific cell surface labeling with an NHS-conjugate (center panel). Staining of the actin cell cortex by phalloidin was used as a reference for cellular localization (right panel). (**B**) Delivery of many newly synthesized (heavy) proteins was increased to the apical surface upon basolateral tannic acid treatment. (**C**) Almost no changes in protein delivery to the basolateral surface proteins were observed upon apical tannic acid treatment. (**D**) Comparison of the heavy-to-light ratio of apical and basolateral surface proteins indicates that some newly synthesized proteins integrate more rapidly into either the apical or basolateral membrane domain, suggesting side-specific turnover rates for these proteins. Blue dots mark proteins that show statistically significant differences between conditions (shown are median values, *n* = 3, two-sided *t*-test FDR = 0.05, s0 = 0.1).

**Figure 5 ijms-23-16193-f005:**
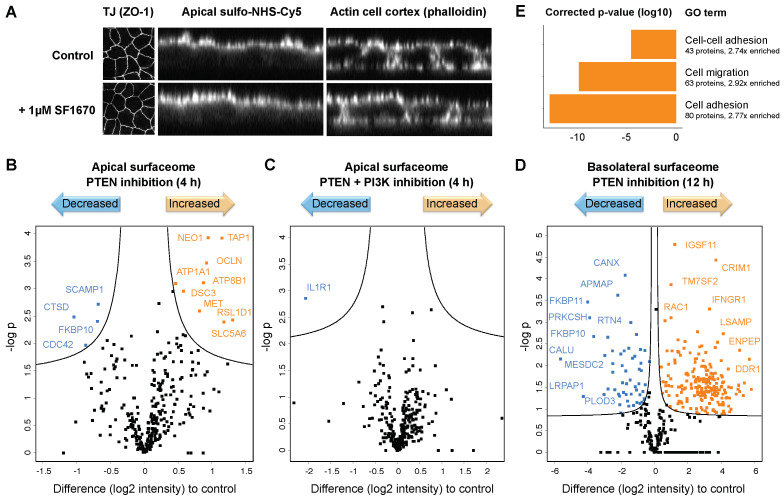
Impaired PTEN function impacts the polarized surfaceome. (**A**) Representative confocal microscopy images of filter-grown MDCK cells showed intact cell layer integrity upon PTEN inhibition by 1 µM SF1670, with no effect on the expression of the tight junction (TJ) protein ZO-1 (left panel) nor on the side-specific cell surface labeling with an NHS-conjugate (center panel). Staining of the actin cell cortex by phalloidin was used as a reference for cellular localization (right panel). (**B**) Label-free quantification of apical surface proteins upon PTEN inhibition by 1 µM SF1670 for 4 h revealed alterations in protein abundance. (**C**) Simultaneous PTEN and PI3-kinase inhibition by 1 µM SF1670 and 1 µM LY294002, respectively, diminished the effect. (**D**) PTEN inhibition caused substantial quantitative changes in the basolateral surfaceome composition. Proteins with a significantly altered abundance compared with the control are marked in orange (increased) and blue (decreased) (*n* = 3, two-sided *t*-test FDR = 0.05, s0 = 0.1). (**E**) Gene ontology analysis of basolateral proteins that showed increased levels upon 12 h of PTEN inhibition revealed enrichment of proteins involved in cell adhesion, cell–cell adhesion, and cell migration (Fisher’s exact test with Bonferroni correction).

**Figure 6 ijms-23-16193-f006:**
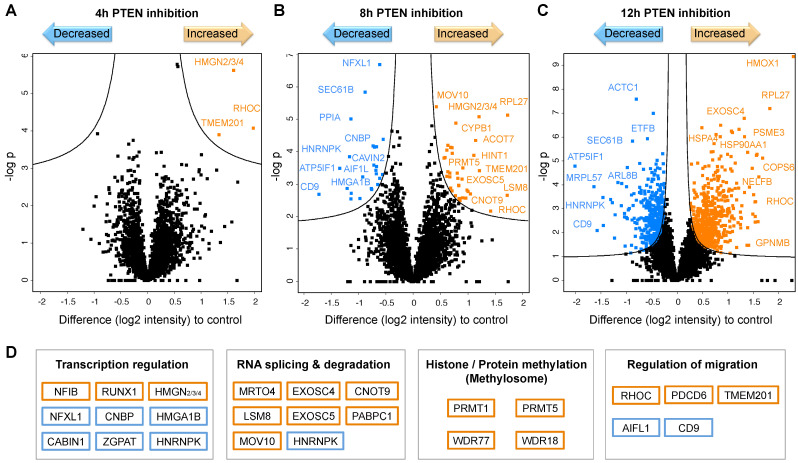
Protein expression changes upon PTEN inhibition in polarized MDCK cells indicate a massive cellular reorganization. (**A**–**C**) Label-free quantification of the cellular proteotype over a time course of PTEN inhibition by 1 µM SF1670 for 4 h, 8 h, and 12 h. (**D**) Functional classes of selected proteins that were found to have an altered abundance after 8 h of PTEN inhibition as potential drivers of proteotype reorganization. Proteins with a significantly altered abundance compared with the control are marked in orange (increased) and blue (decreased) (*n* = 6, two-sided *t*-test FDR = 0.05, s0 = 0.1).

## Data Availability

Mass spectrometry data are available from MASSIVE with the identifier MSV000090530. Analyzed data are available in Appendix A and RAW file annotations can be found in Appendix A.

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
