# Peer review of "Apicobasal Surfaceome Architecture Encodes for Polarized Epithelial Functionality and Depends on Tumor Suppressor PTEN"

_ijms, 2022, doi:10.3390/ijms232416193_

Round 1

Reviewer 1 Report

The manuscript by Annika Koetemann and Bernd Wollscheid describes a quantitative map of the apical vs. basolateral surfaceome of MDCK cells. The manuscript represents a novel and original chemoproteomics approach to identify apical vs. basolateral surface associated proteins by mass spectrometry. In addition, the authors successfully used tannic acid to fix the membrane in a side-specific manner and have shown that PTEN is crucial for polarized localization of cell surface proteins. The manuscript represents a substantial amount of experimental work with sound and verified results and definitely deserves to be published in IJMS. I have only few minor comments:

 1.       To make the study workflow incl. the labeling more clear on the first sight, I suggest to move Fig S1A-B to the manuscript body and improve the description of labeling in Results part of the manuscript for all MS experiments the authors have performed.

2.       The authors have to clearly present in how many biological and technical replicates the MS experiments were performed and define what the term biological/technical replicates replicate means in their setting.

Author Response

Dear Editor Ms. Andjela Stojanovic & dear reviewers,

Please find our revisions attached. We’d like to thank the reviewers for their feedback which we addressed with the detailed point-by-point response below. We addressed all comments carefully by adding text/clarifications, modifying figures, and performing additional statistical testing. Furthermore, MS data of all presented experiments have been uploaded to MassIVE . For further convenience and requested by reviewer 2, protein lists for all experiments including original quantitative data and data on protein identification are accessible now as supplementary excel sheets.

Best regards & we hope to have satisfied the reviewers and look forward to publication of our manuscript in the International Journal of Molecular Science!

Anika Koetemann & Bernd Wollscheid

*****REVIEWER 1*****

https://susy.mdpi.com/user/manuscripts/review/32201770?report=23951054

The manuscript by Anika Koetemann and Bernd Wollscheid describes a quantitative map of the apical vs. basolateral surfaceome of MDCK cells. The manuscript represents a novel and original chemoproteomics approach to identify apical vs. basolateral surface associated proteins by mass spectrometry. In addition, the authors successfully used tannic acid to fix the membrane in a side-specific manner and have shown that PTEN is crucial for polarized localization of cell surface proteins. The manuscript represents a substantial amount of experimental work with sound and verified results and definitely deserves to be published in IJMS. I have only few minor comments:

  1. To make the study workflow incl. the labeling more clear on the first sight, I suggest to move Fig S1A-B to the manuscript body and improve the description of labeling in Results part of the manuscript for all MS experiments the authors have performed.
  • As suggested, Figure S1A-B have been moved to the main body of the manuscript (now Figure 1), including a detailed description of cell surface labeling in the figure legend. Experimental descriptions of cell surface labeling in the MS experiments were extended in the respective Results and Methods (4.5-4.7) parts in addition to the graphical descriptions in Figure 4 for the Tannic acid/polarized protein sorting experiments.
  1. The authors have to clearly present in how many biological and technical replicates the MS experiments were performed and define what the term biological/technical replicates replicate means in their setting.
  • In addition to the statements of replicate numbers (n) in the figure legends, a definition for these numbers has been added to the method section 4.10.

Reviewer 2 Report

Based on a MDCK cell model, the authors study the distribution of apical and basolateral surface proteins using a chemo-proteomics approach. Here, they combined SILAC labelling for quantification and a transwell system to specifically biotinylate apical or basolateral surface proteins followed by enrichment and mass spectrometric analysis. The authors put their results into a functional context, used pulse-SILAC experiment to elegantly investigate intracellular protein trafficking to polarized membrane compartments and furthermore investigated the consequences of PTEN inhibition on the apicobasal surfaceome. An exemplary validation of results with alternative methods like ICC has not been carried out.

Overall, the manuscript is in a very good shape. The used methods are state of the art and the experiments have very well been performed and interpreted. Even if the idea to investigate MDCK cell polarization with chemo-proteomics is not new (Caceres et al. 2019), the current study provides a bigger depth of the results and provides additional data on polarized protein membrane trafficking and the influence of PTEN on membrane polarization. As not many global studies on cell polarization in a comparable quality have been published so far, the manuscript is an outstanding contribution to the field.

Nevertheless, I have some minor comments, which I would suggest to work on before final publication. Especially, I’d suggest to make the quantitative protein data more easily available and to add some methodical details.

Results 2.1 Please provide the information what the abbreviation MDCK stands for.

Supplemental Figure 1 C: Please provide the information what the abbreviation CANLF stands for.

Figure1: Please provide a higher resolution figure.

Results 2.3: I’d suggest to provide some information of how many proteins have been quantified/went into the analysis to get an impression here.

Figure 3a: Is there any reason that basolateral sulfo-NHS-Cy5 staining in tannic acid treated cells is not shown as control?

Figure 3b: Overall, my feeling is that the data is convincing. Nevertheless, I’d find a control interesting in which the Tannic acid treated site is biotinylated to get some insights into the specificity of the approach, but this is mandatory. How it the data distributed? Is the used test appropriate?

Manuscript organization: is 2.3.1 really a subchapter of 2.3 or would 2.4.1 more appropriate?

Discussion: The authors mentioned a comparison with Caceres et al. 2019. Is stated that “The data are remarkably consistent with our map of the apicobasal surfaceome.” I would find it nice if this statement could be supported by some data.

Methods 4.7: Please specify when exactly in the procedure SILAC label samples were pooled. Before or after cell lysis?

Methods 4.7: “AmBic”: I’d suggest to use IUPAC nomenclature for chemicals. What is the composition of “StimLys“ buffer?

Methods 4.8/4.9: It is mentioned that certain peptide amounts were loaded for LC/MS. How and at which timepoint were they quantified? How have non-enriched samples been prepared?

Methods 4.10. How many valid values needed to be present for label-free based quantitative analysis. How was the data distributed before statistical testing? Was it somehow transformed? Were the used statistical tests appropriate for these distributions. Probably SAM has been used for FDR based cut-off determination. This should be mentioned.

It is stated that the MS and search data has been uploaded to the MASSIVE repository. As reviewer login has been provided, it was not possible for me to evaluate the data and to check which data has been uploaded.

A list of the initial apicobasal surfaceome can be found in the supplemental information as graphical item. Nevertheless, there is no data on protein identification and sample specific quantification. In the light of the main findings which basically are catalogues of proteins, I’d strongly suggest to make the protein lists for all experiments including original quantitative data and data on protein identification easily accessible for example as excel sheets as supplemental information.

Author Response

Dear Editor Ms. Andjela Stojanovic & dear reviewers,

Please find our revisions attached. We’d like to thank the reviewers for their feedback which we addressed with the detailed point-by-point response below. We addressed all comments carefully by adding text/clarifications, modifying figures, and performing additional statistical testing. Furthermore, MS data of all presented experiments have been uploaded to MassIVE . For further convenience and requested by reviewer 2, protein lists for all experiments including original quantitative data and data on protein identification are accessible now as supplementary excel sheets.

Best regards & we hope to have satisfied the reviewers and look forward to publication of our manuscript in the International Journal of Molecular Science!

Anika Koetemann & Bernd Wollscheid

******REVIEWER 2******

https://susy.mdpi.com/user/manuscripts/review/32401542?report=24108984

Based on a MDCK cell model, the authors study the distribution of apical and basolateral surface proteins using a chemo-proteomics approach. Here, they combined SILAC labeling for quantification and a transwell system to specifically biotinylate apical or basolateral surface proteins followed by enrichment and mass spectrometric analysis. The authors put their results into a functional context, used pulse-SILAC experiment to elegantly investigate intracellular protein trafficking to polarized membrane compartments and furthermore investigated the consequences of PTEN inhibition on the apicobasal surfaceome. An exemplary validation of results with alternative methods like ICC has not been carried out.

Overall, the manuscript is in a very good shape. The used methods are state of the art and the experiments have very well been performed and interpreted. Even if the idea to investigate MDCK cell polarization with chemo-proteomics is not new (Caceres et al. 2019), the current study provides a bigger depth of the results and provides additional data on polarized protein membrane trafficking and the influence of PTEN on membrane polarization. As not many global studies on cell polarization in a comparable quality have been published so far, the manuscript is an outstanding contribution to the field.

Nevertheless, I have some minor comments, which I would suggest to work on before final publication. Especially, I’d suggest to make the quantitative protein data more easily available and to add some methodical details.

  • Results 2.1 Please provide the information what the abbreviation MDCK stands for.

           Respective information has been added to Results section 2.1.

  • Supplemental Figure 1 C: Please provide the information what the abbreviation CANLF stands for.

Respective information has been added to the figure legend.

  • Figure1: Please provide a higher resolution figure.

Figure 1 has been replaced by a version of higher resolution and will be provided in .pdf format during the final manuscript submission.

  • Results 2.3: I’d suggest to provide some information of how many proteins have been quantified/went into the analysis to get an impression here.

Information on the numbers of quantified proteins in this experiment have been added to subsection 2.3.

  • Figure 3a: Is there any reason that basolateral sulfo-NHS-Cy5 staining in tannic acid treated cells is not shown as control?

This experiment is meant as a control for the impact of tannic acid on the integrity of the cell layer (side-specificity of NHS-labeling is demonstrated in figure S1). If tannic acid was affecting cell layer integrity, it would be expected to do so no matter from which side it is applied to the cells. Thus, the absence of apically applied sulfo-NHS-Cy5 on the basolateral cell surface indicates that tannic acid does not have any impact on cell layer integrity.

  • Figure 3b: Overall, my feeling is that the data is convincing. Nevertheless, I’d find a control interesting in which the Tannic acid treated site is biotinylated to get some insights into the specificity of the approach, but this is mandatory. How is the data distributed? Is the used test appropriate?

Since tannic acid is acting as a fixative of the treated membrane domain, it is not compatible with cell surface protein biotinylation or downstream MS analysis - therefore a control in which the tannic acid treated site is biotinylated has not been included.

Based on the reviewer’s comment we have had another look at the data of this experiment and adapted our analysis strategy: Since it is a pulsed SILAC experiment at an early time point and we are looking at heavy-to-light ratios, the data is a bit skewed (or rather the left end of the normal distribution is cut off) as many proteins have very low or no heavy signals yet. Therefore statistical testing has now been performed on the log-transformed ratio values, as these show a more normal distribution. Although trends and interpretation of the data remain the same, figures that show significant hits marked in blue have been replaced by an updated version (for visualization the log values have been transformed back as it is more “visual”/intuitive for the reader). The method section has been updated regarding data analysis for this experiment.

  • Manuscript organization: is 2.3.1 really a subchapter of 2.3 or would 2.4.1 more appropriate?

Sub-sectioning has been adjusted as suggested.

  • Discussion: The authors mentioned a comparison with Caceres et al. 2019. Is stated that “The data are remarkably consistent with our map of the apicobasal surfaceome.” I would find it nice if this statement could be supported by some data.

A comparison of the apicobasal MS quantification of selected proteins, which have been verified in the study by Caceres et al. by immunoblotting, as well as a calculation of the overall difference in the apicobasal quantification across all proteins that were detected in both studies, have been added to the Discussion section.

  • Methods 4.7: Please specify when exactly in the procedure SILAC label samples were pooled. Before or after cell lysis?

Information about sample pooling after cell lysis has been added to the method section (4.7).

  • Methods 4.7: “AmBic”: I’d suggest to use IUPAC nomenclature for chemicals. What is the composition of “StimLys“ buffer?

“AmBic” has been replaced by NH4HCO3 and the composition of the “StimLys” buffer has been added to the method section.

  • Methods 4.8/4.9: It is mentioned that certain peptide amounts were loaded for LC/MS. How and at which timepoint were they quantified? How have non-enriched samples been prepared?

Additional details about peptide quantification and sample processing of non-enriched samples have been added to sections 4.7, 4.8 and 4.9.

  • Methods 4.10. How many valid values needed to be present for label-free based quantitative analysis. How was the data distributed before statistical testing? Was it somehow transformed? Were the used statistical tests appropriate for these distributions. Probably SAM has been used for FDR based cut-off determination. This should be mentioned.

Analysis and statistical testing of label-free data, which showed a normal distribution, have been performed using the Perseus software in a standard manner for the generation of volcano plots. More detailed information about data processing, including log-transformation, two-sided t-test and permutation-based FDR calculation (appropriate for normally distributed data), as well as the minimum number of valid values (2) for proteins to be included in further analysis has been added to the method section.

  • It is stated that the MS and search data has been uploaded to the MASSIVE repository. As reviewer login has been provided, it was not possible for me to evaluate the data and to check which data has been uploaded.

MS data of all presented experiments have been uploaded to MassIVE (Reviewer access: ftp://[email protected], PWD:wlab@2022). A list with sample annotations for all uploaded files has been added to the Supplements (Table S 3).

  • A list of the initial apicobasal surfaceome can be found in the supplemental information as graphical item. Nevertheless, there is no data on protein identification and sample specific quantification. In the light of the main findings which basically are catalogues of proteins, I’d strongly suggest to make the protein lists for all experiments including original quantitative data and data on protein identification easily accessible for example as excel sheets as supplemental information.

Protein identification and sample specific quantification data has been added as Excel file to the supplements (Table S2). Please note that the Table on apicobasal protein quantification (Supplementary Table S1) already contains quantification data of the individual replicates (SILAC forward (fw) experiments 1 and 2; SILAC reverse (rv) experiments 1 and 2), where “NaN” represents non-detected/quantified proteins in a specific replicate. The table legend has been revised to make this information more accessible to the reader.